# Patient-generated data in the management of HIV: a scoping review

Clara Hewitt ID,[1] Karen C Lloyd ID,[1] Shema Tariq ID,[1] Abigail Durrant ID,[2] Caroline Claisse ID,[2] Bakita Kasadha ID,[3] Jo Gibbs ID[1]

[1]Institute for Global Health, University College London, London, UK
[2]Open Lab, School of Computing, Newcastle University, Newcastle upon Tyne, UK
[3]Terrence Higgins Trust, London, UK

**Correspondence to**
Dr Karen C Lloyd;
k.lloyd@ucl.ac.uk

## ABSTRACT

**Objectives** Patient-generated data (PGData) are an emergent research area and may improve HIV care. The objectives of this scoping review were to synthesise, evaluate and make recommendations based on the available literature regarding PGData use in HIV care.

**Design** Scoping review.

**Data sources** Embase, Medline, CINAHL Plus, Web of Science, Scopus, PsycINFO and Emcare databases.

**Eligibility criteria** Studies involving PGData use within HIV care for people living with HIV and/or healthcare professionals (HCPs) published before February 2021.

**Data extraction and synthesis** Data were extracted using a table and the Mixed Methods Appraisal Tool was used to assess empirical rigour. We used thematic analysis to evaluate content.

**Results** 11 articles met the eligibility criteria. Studies were observational, predominantly concerned hypothetical or novel digital platforms, mainly conducted in high-income settings, and had small sample sizes (range=10–160). There were multiple definitions of PGData. In the majority of studies (n=9), participants were people living with HIV, with a few studies including HCPs, informatics specialists or mixed participant groups. Participants living with HIV were aged 23–78 years, mostly men, of diverse ethnicities, and had low educational, health literacy and income levels.

We identified four key themes: (1) Perceptions of PGData and associated digital platforms; (2) Opportunities; (3) Anticipated barriers and (4) Potential impact on patient–HCP relationships.

**Conclusions** Use of PGData within HIV care warrants further study, especially with regard to digital inequalities, data privacy and security. There is a need for longitudinal data on use within HIV in a variety of settings with a broad range of users, including impact on clinical outcomes. This will allow greater understanding of the role of PGData use in improving the health and well-being of people living with HIV, which is increasingly pertinent as digital healthcare becomes more widespread as a result of COVID-19.

## BACKGROUND

HIV remains a major global health challenge. Antiretroviral therapy (ART) has extended the life expectancy of people living with HIV, transforming HIV into a chronic condition.[1] Effective HIV care requires both long-term clinical follow-up as well as high levels

### Strengths and limitations of this study

► This scoping review provides an overview and synthesis of the emergent literature of patient-generated data use within HIV care, providing a foundation of understanding to inform future research.
► The Mixed Methods Appraisal Tool allowed assessment of mixed study designs.
► Literature searching yielded a limited number of articles, with most articles obtained via manual referencing.
► Only English studies were reviewed, which may bias and thus narrow the scope of the findings to English-speaking settings.

of adherence to ART. This facilitates viral suppression, reducing HIV-related morbidity and mortality, as well as preventing onward HIV transmission.[2] However, HIV care is often complex; people living with HIV have a greater risk of comorbid conditions such as poor mental health and cardiovascular disease compared with their HIV-negative counterparts.[3 4] These comorbidities (often a result of the virus, side effects of ART, and/or complex socioeconomic factors) may require considerable input from a wide range of healthcare professionals (HCPs), which has a consequential impact on healthcare resources, even in high-income settings.[1 5] With health systems already financially and time poor, it is important to identify innovative approaches to HIV care that can support quality of life, while addressing the challenges posed by increasing comorbidities and an ageing population. Digital technologies present opportunities through which to improve HIV care, and in turn, the health and quality of life of people living with HIV.[6]

The role of self-management in the care of long-term conditions such as HIV is becoming increasingly important.[6] The ubiquity of digital devices and applications used for health purposes can support self-management and has created fertile grounds for patient-generated data (PGData)

collection.[7][8] Although no universal PGData definition exists, multiple authors across the literature have adopted the broad definition[9] that PGData constitute any health-related data which are created or collected by patients or designated proxies to address a health concern.[7][10–14] As such, PGData include medication dosages, physical activity, dietary intake, sleep and mood patterns, and can be collected in different modalities including text, pictures, voice or video recordings, and numerical information, for example, questionnaire scores or physiological measurements.[11][15] PGData allow more frequent, remote and longitudinal tracking of multiple types of health information, and can be rapidly shared between patients and HCPs, or between patients themselves, usually using wireless connections.[13][16] Additionally, PGData can serve as an adjunct to health information communicated during clinical consultations; they can demonstrate baseline health measures bespoke to each individual, which may ultimately facilitate more patient-centred, personalised care.[8][14][17] Consequently, PGData allow greater real-time insight into health fluctuations or anomalies relative to a baseline, permitting holistic appreciation of patient health for both patients and their HCPs, and potentially enabling more appropriate and timely interventions.[5][9–11][15][16][18]

While the use of PGData for the management of other long-term conditions such as cancer[14] and diabetes[18] has been investigated, to our knowledge, no review of PGData use within HIV care has been conducted to date. This scoping review is timely, given that COVID-19 has led to rapidly increased adoption of remote healthcare and telemedicine practices within clinical care, which will likely remain in place beyond the current pandemic.

## Objectives

This scoping review aimed to (1) synthesise and evaluate the existing emergent literature on PGData use within HIV care; (2) identify the opportunities and challenges it presents; and (3) consider the impact of PGData use on people living with HIV and their HCPs. For the purposes of this review, we interpreted 'HIV care' as being synonymous with HIV management, and to mean all HIV-related healthcare from the point of diagnosis.

## METHODS

We conducted a scoping review of literature on PGData use within HIV care. We adopted this approach to enable a broad focus and preliminary synthesis of available evidence[19] in this new, emerging research area. We have reported the review in accordance with the Preferred Reporting Items for Systematic Reviews and Meta-analysis extension for Scoping Reviews guidelines[20] (see online supplemental appendix A). A review protocol has not been registered.

---

**Box 1  Summary of inclusion and exclusion criteria**

**Inclusion criteria**
► Studies which considered the perspectives of healthcare professionals or people living with HIV regarding patient-generated data (PGData) use within HIV management.
► Studies where PGData use met the definition adopted.[10]
► Where studies involved patients, these needed to be people living with HIV.
► PGData collection by people living with HIV had to be conducted via a digital platform.
► PGData collected in relation to living with HIV.
► Any PGData created had to be collected independently by people living with HIV, and not in response to a reminder or prompt.

**Exclusion criteria**
► Studies involving PGData which were not generated independently by people living with HIV, or where PGData were created in response to the receipt of a reminder or prompt.
► Studies concerning PGData use in the diagnosis, prevention or risk reduction of HIV.
► Studies solely involving a two-way communication system, including studies involving digital messaging or forum platforms.
► Conference proceedings.

## Search strategy

During July 2019, we searched Medline, Embase, Emcare, CINAHL, PsycINFO, Scopus and Web of Science databases for articles. Additional grey literature databases were searched using the same strategy: National Institute of Clinical Excellence: Health and Social Care, TRIP medical database, ProQuest Dissertations and Theses, OpenAIRE, Bielefeld Academic Search Engine, OpenDOAR, Open Access Dissertations and Theses, DART-Europe, Open Grey and Semantic Scholar. Searches were restricted to English language and the terms used were adapted to meet the requirements of each database. Detailed search terms can be found in online supplemental appendix B. Article reference lists were also manually searched to obtain additional literature which the formal search strategy may have omitted.

CH conducted the original screenings in conjunction with JG and KCL. ST and CH reran literature searches of all databases in February 2021 using the same search strategy. No further literature within scope was identified.

## Eligibility criteria

Study inclusion was contingent on the terms outlined in box 1 and temporal limits were not imposed to keep the scope of literature generated as broad as possible given the infancy of the literature base.

We chose to exclude PGData which are collected in response to prompts from external sources, for example, mobile application (app) or short message service reminders. While PGData prompts are not specifically addressed in the PGData definition we employed, we made this exclusion on the basis that we believed PGData collected following prompts are not truly 'generated' autonomously by patients. We also employed this

exclusion based on our consideration that prompted PGData may mean an individual has a qualitatively different experience of PGData collection and use, and that any prompted PGData collected may not be a true reflection of natural, uninfluenced patient engagement with PGData, for example, the PGData which individuals would naturally collect unprompted.

### Selection of sources of evidence

The retrieved articles were exported into Mendeley Reference Manager and duplicates were removed. Titles and abstracts were screened, and full texts were assessed using the eligibility criteria. A data extraction table (see online supplemental appendix C) was used to summarise data from articles meeting the eligibility criteria for analysis.

### Empirical appraisal and thematic analysis of included articles

Due to heterogeneity of study designs, empirical quality was assessed using the Mixed Methods Appraisal Tool (MMAT,[21] see online supplemental appendix D). Thematic analysis[22] was conducted based on the qualitative findings from the identified articles. We employed an inductive approach due to the nascent nature of this research field.

For the thematic analysis, primarily individual article-specific mind maps were created to highlight key concepts identified per article.[23] Subsequently, a comprehensive mind map was created to synthesise key topics across all 11 reviewed articles. Topics recurring twice or more between articles were highlighted as having thematic importance and these were finally organised into four overarching themes and three subthemes. Themes were discussed by CH, KCL, JG and ST until consensus was reached.

### Patient and public involvement

This manuscript reports on a review of the literature. As such, no patients or members of the public were explicitly involved in the scoping process, however, one coauthor is a peer-researcher (with lived experience of HIV) within a third sector organisation.

## RESULTS

### Search results

Original literature searches were conducted in June 2019. A total of 2353 articles were identified through academic databases and grey literature keyword searching (figure 1). Seventy-three full-text articles were assessed, of which five fulfilled the eligibility criteria for inclusion. A further six eligible studies were identified from a manual search of reference lists. A total of 11 articles were

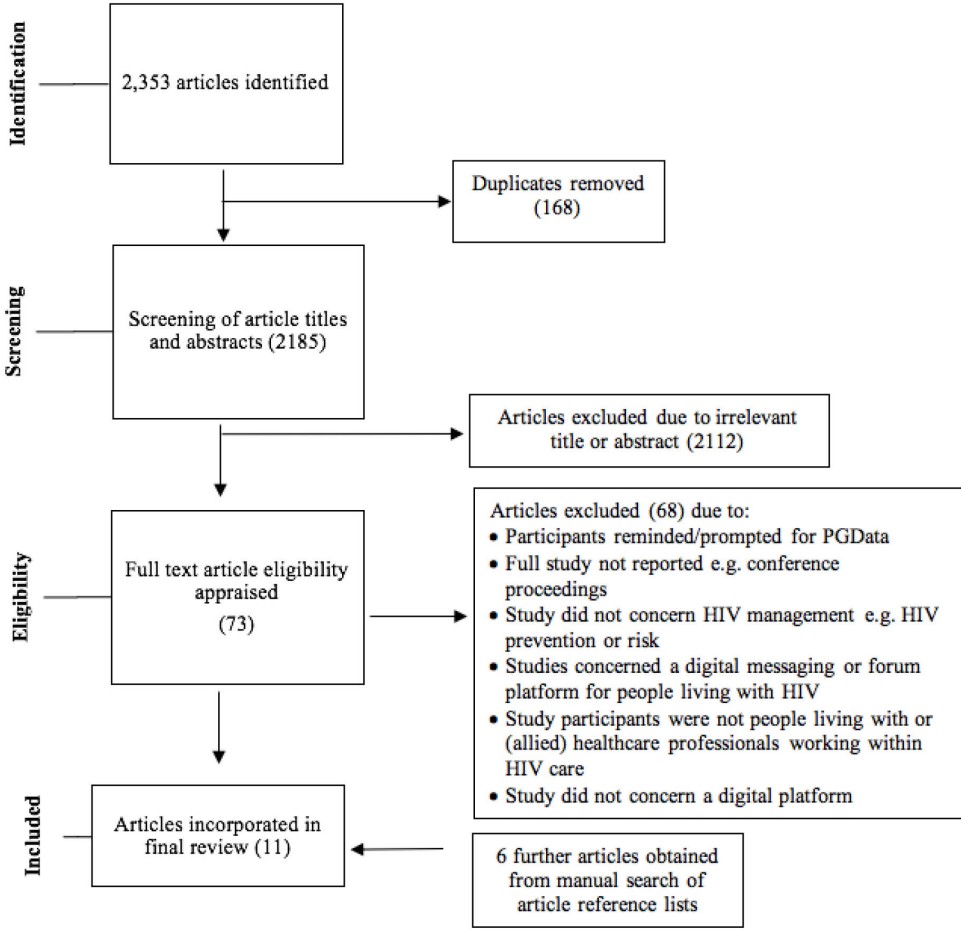

**Figure 1**  PRISMA flow diagram. PGData, patient-generated data; PRISMA, Preferred Reporting Items for Systematic Reviews and Meta-Analyses.

included for final review (see article summaries, table 1). An updated search conducted in February 2021 did not identify any further eligible studies.

## Participants and demographics

Sample sizes of included studies ranged from 10 to 160 participants. Participants were predominantly people living with HIV, however, some studies involved HCPs, allied HCPs, informatics specialists or mixed participant groups (see online supplemental appendix C). Apart from one study,[24] where nurses and doctors were participants, role specialisms were not identified. Where reported, studies primarily included men who have sex with men and heterosexual men and women. The age range of people living with HIV across the studies was 23–78 years. Participants were mostly men and of black (including black African and African American) ethnicity. Only two studies involved transgender people living with HIV, despite this being an HIV key population. Where reported, people living with HIV were most often from low-income, low-educational backgrounds.

## Study characteristics

The empirical quality of the 11 articles reviewed was varied (see MMAT assessment, online supplemental appendix D). Studies were predominantly conducted in the USA (eight studies),[24–31] with others based in China,[32] England[33] and other parts of Europe.[34] All PGData platforms were mobile or web-based except for one study involving a medication storage device.[32] Prior to analysis, articles were grouped into three categories according to the type of digital platform used: hypothetical, prototype or pre-established (see table 1) in order to distinguish between the different natures of digital platform types. Hypothetical platforms were those where researchers had developed a PGData platform concept, with no tangible product in existence (five studies),[24 27 30 33 34] prototype platforms involved PGData platforms being piloted (three studies)[28 29 31] and pre-established platforms referred to PGData platforms already in existence which were being trialled (three studies).[25 26 32] Study designs included user testing,[25 28 29 32] focus groups[26 30 31] and semistructured interview sessions.[24 33 34] One study also included patient–HCP co-design workshops.[34]

User testing studies varied in structure; two studies trialled beta mobile app prototypes for PGData collection where people living with HIV and informaticians completed tasks while simultaneously voicing feedback,[28 29] while others examined the impact of medication storage device use[32] and a 6-week personal health record app training programme for people living with HIV.[25] User testing studies focused on the perceived usability and feasibility of PGData collection and associated digital platforms; some studies employed quantitative assessments to measure these[25 28 29] (see online supplemental appendix C). No studies conducted user testing with HCPs. Studies conducting interviews and focus group discussions tended to use open-ended questions or follow

discussion guides, and centred around issues of PGData platform acceptability, feasibility, and anticipated barriers and facilitators to PGData adoption.

## Overview of key themes

Our analysis highlighted four primary cross-cutting themes and three subthemes, including: perceived acceptability, feasibility and usability of PGData; opportunities presented by PGData; anticipated barriers to PGData use; and how PGData may impact HIV care and HCP–patient relationships.

## Acceptability, feasibility and usability of digital platforms for PGData

Participants' views regarding the acceptability, feasibility and usability of PGData differed. Some people living with HIV were receptive to digitally tracking their PGData, though this was contingent on data security.[30] Widespread smartphone access, absence of PGData platform cost, clear design and easy navigability of digital PGData platforms increased the perceived feasibility of PGData use.[24 26 32 33]

Perceptions regarding the perceived suitability of PGData use for people living with HIV varied; some HCPs suggested PGData use would be better suited to newly diagnosed individuals, those with multimorbidities or higher education levels,[24 34] while others proposed newly diagnosed people living with HIV may be unsuitable.[34] Some HCPs questioned whether individual differences including motivation may affect PGData collection and engagement.[24] No data from people living with HIV regarding suitability were reported.

## Opportunities presented by PGData for HIV care

Participants cited several perceived benefits of PGData use within HIV care, including reduced financial and time burdens,[31] convenience and increased health proactivity.[26 31 33] Some people living with HIV perceived PGData collection could increase their medication adherence, perceived control, understanding and engagement surrounding their HIV care[33] due to their awareness of ongoing PGData monitoring.[24 30 32]

## Anticipated barriers and challenges associated with PGData

Across the studies, several perceived challenges to PGData use emerged, including privacy and security, accessibility and reliability of PGData, and impact on HCPs.

### Privacy and security

Participants expressed substantial privacy and security concerns regarding PGData collection and analysis,[25 26 30 31] which some authors attributed to persisting HIV stigma.[31 32 34] Participants voiced concerns that poor security and privacy may result in unwanted disclosures of sensitive data including HIV status, although how this might occur was not explicitly articulated.[30–32 34] Several people living with HIV emphasised desires for security mechanisms including encryption,[30 33 34] and some people

| Author(s) | Participants, design, location, study length | Digital platform | Overview |
|---|---|---|---|
| **Table 1** | Summary of included articles | | |
| **Studies concerning hypothetical digital platforms** | | | |
| Bussone[33] | N=16 people living with HIV<br>Qualitative, individual semistructured interviews of approximately 30 min<br>England (UK)<br>Study length unspecified | No specific digital platform used | Aimed to understand how people living with HIV monitor their personal health data. Addressed topics including participants' current self-management behaviours and issues surrounding current, previous or desired means of tracking personal health information. |
| Marent et al[34] | N=160 participants<br>▶ 97 people living with HIV<br>▶ 63 HCPs working in HIV care<br> – 40 doctors<br> – 10 nurses<br> – 4 psychologists<br> – 4 pharmacists<br> – 2 social workers<br> – 2 nutritionists<br>▶ 1 sexologist<br><br>Qualitative, 14 recorded workshops and 22 semistructured interviews<br>Brighton (UK), Lisbon (Portugal), Barcelona (Spain), Antwerp (Belgium) and Zagreb (Croatia)<br>January–June 2016 | No specific digital platform used | Aimed to determine perceptions of, and advantages and concerns surrounding, the development of a mobile app for HIV management. Addressed topics including perceived challenges and barriers to the use of a mobile app for HIV management, current mobile app usage and perceived useful features of a mobile app for HIV management. |
| Nokes et al[27] | N=100 people living with HIV or AIDS<br>Quantitative<br>New York, USA<br>Study length unspecified | No specific digital platform used | Aimed to assess self-efficacy of people living with HIV regarding use of a hypothetical personal health record. Participants received an explanation of a personal health record and completed questionnaires measuring their self-efficacy in relation to digital and paper-based personal health records, condom use and chronic disease management. |
| Ramanathan et al[30] | N=29 people living with HIV<br>Qualitative: focus group discussions<br>Los Angeles, USA<br>Study length unspecified | No specific digital platform used | Aimed to assess feature preferences for a mobile app for HIV self-management. Addressed topics including privacy, goal-setting, data-capturing methods, feedback regarding app-user behaviours and the role of reminders. |
| Swendeman et al[24] | N=12 HCPs working in HIV care<br>▶ 3 doctors<br>▶ 5 nurses<br>▶ 2 psychosocial case workers<br>▶ 1 outreach worker<br>▶ 1 psychotherapist<br>Qualitative, individual 45–60 min semistructured interviews<br>Los Angeles, USA<br>Study length unspecified | No specific digital platform used | Aimed to assess HCP attitudes regarding a hypothetical mobile app (for people living with HIV) and online dashboards (for HCPs) for HIV management. Addressed topics including usability, acceptability and barriers to digital platform use. |
| **Studies concerning prototype digital platforms** | | | |
| Schnall et al[31] | N=37 case managers working with people living with HIV<br>Qualitative, 1-hour focus group discussions<br>New York, USA<br>March–December 2008 | SelectHealth Continuity of Care Document, includes key health information including latest medical test results, pharmaceutical record, visits to healthcare facilities and patient contact information | Aimed to determine factors influencing usage and acceptability of online 'SelectHealth Continuity of Care Document' for people living with HIV, before the platform's release. Addressed topics including facilitators and barriers to the use of the continuity of care record. |

Continued

**Table 1** Continued

| Author(s) | Participants, design, location, study length | Digital platform | Overview |
|---|---|---|---|
| Schnall et al[28] | N=15<br>► 10 people living with HIV<br>► 5 informaticians<br>Mixed: quantitative,qualitative<br>New York, USA<br>Study length unspecified | No description of app prototype provided | Aimed to test a prototype mobile app designed to facilitate HIV management in people living with HIV. Participants received a description of a mobile app prototype for HIV management. Informaticians tested the model for 45–90 min and completed a Heuristic Evaluation Checklist. People living with HIV evaluated prototype app screenings and completed the Post Study System Usability Questionnaire. |
| Stonbraker et al[29] | N=25<br>► 20 people living with HIV who possessed at least one HIV-associated non-AIDS condition<br>► 5 informaticians<br>Mixed: quantitative; qualitative<br>New York, USA | Novel mobile app 'VIP-HANA' (Video Information Provider for HIV-associated non-AIDS conditions) | Aimed to determine usability of novel mobile app 'VIP-HANA' designed to aid people living with HIV with management of HIV/non-HIV-related symptoms, and to determine where improvements to the app were needed. After receiving descriptions of the app, participants were assigned tasks on a beta version of the app and had to describe aloud their thoughts and actions while their activities were tracked. People living with HIV completed one questionnaire to assess their health literacy level, and two questionnaires relating to the app's usability. |
| **Studies concerning pre-established digital platforms** | | | |
| DeSilva et al[32] | N=10 people living with HIV who used injection drugs<br>Individual qualitative interviews<br>Nanning, China<br>Study length unspecified | Wisepill medication storage container; once opened a wireless signal immediately transmits to a server, the data of which can be obtained by HCPs | Aimed to determine the feasibility and acceptability of using a 'Wisepill' device in injecting drug users who were HIV positive. Participants used the 'Wisepill' device to monitor their antiretroviral medication adherence for 1 month. Addressed topics including acceptability, feasibility and usability of the device. |
| Luque et al[25] | N=29 people living with HIV<br>Quantitative<br>Rochester, New York, USA<br>6 weeks | MyMedical app, (downloaded from iTunes store) | Aimed to determine acceptability and usability of using the MyMedical app as a personal health record on an iPod Touch device, and to observe the effect of using a digital personal health record on HIV treatment self-efficacy. Participants underwent six 90 min training sessions and were given 'homework' tasks to rehearse between training sessions. |
| Odlum et al[26] | N=57 people living with HIV<br>► Quantitative: 42 people living with HIV<br>► Qualitative: 15 people living with HIV; 8 users of the MyHealthProfile app, 7 non-users of the MyHealthProfile app<br>Mixed: quantitative, qualitative<br>New York, USA<br>Study length unspecified | MyHealthProfile app acts as a continuity of care record for people living with HIV to track their personal health record via an internet connection | Aimed to determine usability of the MyHealthProfile app and to determine where improvements could be made to inform a new version of the app named MyHealthProfile-plus. Participants completed two surveys relating to the usefulness and content of the pre-established MyHealthProfile app. Participants completed 60 min focus groups regarding the development of new MyHealthProfile-plus app |

HCPs, healthcare professionals.

living with HIV favoured paper-based PGData tracking methods due to digital platform apprehension.[33]

### Accessibility and reliability of digital infrastructure and collected PGData

There were prevalent concerns regarding the reliability of digital platforms and PGData itself. Some studies reported concerns that technological disruptions and financial instability may hinder digital platform accessibility and usage.[24 32] Additionally, some participants highlighted PGData reliability could be impaired by incorrect PGData recording.[31] Some people living with HIV warned of intentional retrospective editing to deliberately manipulate PGData.[30 33]

### Impact on HCPs

HCPs were apprehensive that pre-existing time constraints and large workloads would hinder PGData reviewing and that PGData involvement would only magnify these burdens.[24 34] People living with HIV did not comment on the impact of PGData on HCPs.

## Perceived impacts of PGData on patient–HCP relationships

Participants felt that engaging with PGData within clinical consultations could facilitate a more comprehensive picture of how a person may be managing their HIV.[24 33 34] While people living with HIV desired PGData platforms allowing two-way communication with HCPs, HCPs feared increased workloads.[34]

Some people living with HIV thought that PGData use may provide opportunities to strengthen patient–HCP relationships.[34] However, contrastingly, other participants expressed concern that PGData misinterpretation could impair pre-established face-to-face patient–HCP interactions.[30 34] Both people living with HIV and HCPs believed PGData use could complement, rather than supersede, traditional in-person HIV consultations, although details of such a hybrid model were absent.[34]

## DISCUSSION

This review provides a critical synthesis of the nascent literature on PGData use specifically for HIV care, and highlights several unanswered questions. Across the 11 articles, study designs and methodologies were varied and a spectrum of digital PGData platforms were considered. The views of people living with HIV and HCPs were broadly consistent, apart from the potential impact of PGData on patient–HCP relationships.

### Individual and contextual differences

Our findings suggest that PGData use may not be suitable for all people living with HIV; indeed, perceptions of PGData and its use within HIV care may differ greatly across patients and contexts.[35] For example, while some patients felt reassured by monitoring their long-term health condition, for others, health monitoring could be tiring, intrusive, anxiety-provoking and excessively time-consuming.[9 30 36 37] It is important that digital health interventions to support HIV care and self-management do not further reinforce stigma and use language that is accessible to all users.[38] As the majority of studies did not include participants from key HIV populations, such as women, people who are transgender or people who inject drugs, knowledge of PGData use in these communities remains limited.[15] The current review's bias towards high-income settings further limits its generalisability given the global HIV burden lies predominantly in low-income and middle-income settings.[39] Furthermore, digital inequalities due to limited access and poor digital literacy[40 41] will impact acceptability, feasibility and inequalities in PGData use, with the COVID-19 pandemic starkly highlighting these digital divides.[42 43] Further research is needed to better understand these individual and contextual nuances.

### Impact on patient–HCP relationships

Our findings highlight dichotomies between people living with HIV and HCPs; while people living with HIV appeared to welcome increased HCP contact and HCP-analysed PGData, HCPs expressed concerns around additional workload and expectations.[11 14 16 24 34 44 45] A PGData guidance framework may be necessary to counteract competing priorities between HCPs and patients regarding such communication preferences.[9 12] West *et al* suggest one possible workflow model to address these concerns.[46] PGData use within HIV care may also impact patients' levels of responsibility, autonomy and control over their healthcare relative to HCPs.[37 47] Thus, it seems plausible that PGData will impact relationships between people living with HIV and HCPs. The extent of this impact is presently unknown, but it may vary in its effect on both patients and HCPs, as well as across settings and cultural contexts.

### Barriers to PGData uptake

Across the studies, concerns from both patients and HCPs regarding the privacy, security and reliability of PGData were common, indicating considerable apprehension around the use of PGData. Regarding PGData platforms, in this review mobile apps were the most-used digital platform; however, more broadly, apps have received repeated criticism for lacking regulation and approval for medical use, potentially threatening the quality, validity and reliability of mobile-collected data.[36 48–52] Furthermore, many apps lack privacy policies or data encryption measures, potentially risking data exploitation, theft or unauthorised access.[49 53 54]

Given the prominent PGData privacy and security concerns highlighted in this review, without adequate regulatory, privacy and security protocols, people living with HIV may be less inclined to engage in PGData. Furthermore, both data security mistrust and social desirability, where patients may under-report or over-report actions to appear more socially acceptable, may increase the likelihood of deliberate PGData manipulation, degrading its reliability.[47 53] Ultimately, PGData adoption into HIV care may be jeopardised if concerns regarding the safety, privacy and trustworthiness of the platforms used to collect PGData, or the PGData itself, prevail among people living with HIV and HCPs.[50 54]

### Limitations

#### Terminology and search strategy

The small article yield of 11 articles has several possible explanations. Many articles were out of scope and considered ineligible; inconsistencies in PGData terminology across the literature[55] may have exacerbated this and potentially rendered relevant articles unidentifiable using the current review's search strategy. Excluding conference proceedings, coupled with challenges in locating full-text articles, may mean some studies were overlooked. Consequently, the structured search strategy proved less fruitful in yielding articles relative to manual reference-searching, a finding other authors have observed.[56] Additionally, author repetition across the studies calls into question whether the same, or similarly recruited, samples were used between studies, raising further generalisability concerns.

#### Lack of real-world evidence

The majority of studies were exploratory, involving hypothetical or pilot platforms, which limits their ecological validity

and applicability. Additionally, the heterogeneity of PGData platforms used prevents their direct comparison and evaluation. Although the digital platforms addressed were biased to mobile phones, this reflects the wider digital health literature and this platform is more established than other recent technological developments (eg, wearable devices).[36] Finally, although PGData use may require training, the one training study[25] involved such high training intensity that its real-world feasibility is unlikely.

### Future research recommendations

Based on this review, the following four areas warrant further research:

#### Real-world and longitudinal evidence

Resulting from the absence of real-world, longitudinal evidence of PGData use in HIV care, a full-scale pragmatic trial is necessary to enhance the research's ecological validity and enable greater insight into practical issues surrounding PGData usage within HIV care. The impact of any of the PGData platform categories we identified—pilot, hypothetical or novel—could be explored in trial settings. Additionally, a greater understanding of how PGData use may fluctuate over time is needed.

#### Impact of PGData use on clinical outcomes

The direct relationship between PGData use and clinical outcomes in people living with HIV, such as medication adherence, quality of life or mental well-being, remains unknown and requires investigation.

#### Digital platforms and PGData types

The effectiveness and optimal digital platform type(s) for PGData collection and monitoring require evaluation. Additionally, exploration of the most useful types of PGData to track is needed.

#### Different contexts and HIV populations

Research exploring PGData use within HIV care in more diverse populations of people living with HIV is critically needed to increase the literature's representativeness for HIV key populations and low-income and middle-income settings. Future research must improve the scope of geographical and patient contexts to avoid further magnifying research inequities regarding PGData use within HIV care.[5 57]

### CONCLUSIONS

The emerging PGData field presents opportunities to improve patient self-management and personalised care, which are both increasingly needed due to health system resource constraints and the complexities of HIV care. In recent months, HIV care, services and people living with HIV worldwide have been negatively impacted by the COVID-19 pandemic, causing decreased HIV testing, disrupted antiretroviral treatment procurement and poorer psychological well-being of people living with HIV.[58 59] However, the pandemic has highlighted the

value of, and significant shift towards, digital healthcare delivery,[60 61] which may provide expedited leverage for further PGData research. Going forwards, the real-world applicability of PGData within HIV care must be elucidated, including navigating the heterogeneity and optimal suitability of PGData platforms for people living with HIV within different contexts. Additionally, PGData integration within HIV care will be conditional on adequately addressing outstanding privacy, data security, reliability and HCP–patient relationship concerns prior to implementation.[14 62] Overall, PGData may hold the potential to transform HIV care, in turn realising improvements in supporting not only people living with HIV but also the wider health systems in which this care is provided.

**Collaborators** Not applicable.

**Contributors** CH, KCL, ST, AD, CC, BK and JG contributed to the conception and/or design of the study. CH led, and KCL and JG contributed to the establishment of the search strategy and method of analysis. Article reviewing and data analysis were led by CH with contribution from KCL and JG. CH drafted the original manuscript with supervision and guidance from JG, KCL and ST. All authors (CC, KCL, ST, AD, CC, BK and JG) have made contributions to the drafting and revising of the article, and have approved the final version.

**Funding** This research received no specific grant from any funding agency in the public, commercial or not-for-profit sectors. All authors (with the exception of CH) receive research funding from the INTUIT (Interaction Design for Trusted Sharing of Personal Health Data to Live Well with HIV) research project, which is funded by the Engineering and Physical Sciences Research Council (EP/R033900/1).

**Competing interests** None declared.

**Patient consent for publication** Not required.

**Provenance and peer review** Not commissioned; externally peer reviewed.

**Data availability statement** All data relevant to the study are included in the article or uploaded as supplemental information. This manuscript reports on a scoping review. All articles included in the sample are included in the article and/or as supplemental material.

**ORCID iDs**
Clara Hewitt http://orcid.org/0000-0002-5711-0716
Karen C Lloyd http://orcid.org/0000-0002-6310-6836
Shema Tariq http://orcid.org/0000-0001-9802-7727
Abigail Durrant http://orcid.org/0000-0003-4752-5376
Caroline Claisse http://orcid.org/0000-0003-4002-6136
Bakita Kasadha http://orcid.org/0000-0003-2862-183X
Jo Gibbs http://orcid.org/0000-0001-5696-0260

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
