## [Reviewer comments · BMJ Open]

ARTICLE DETAILS

TITLE (PROVISIONAL)	Patient-generated data in the management of HIV: a scoping review
AUTHORS	Hewitt, Clara; Lloyd, Karen; Tariq, Shema; Durrant, Abigail; Claisse, Caroline; Kasadha, Bakita; Gibbs, Jo

VERSION 1 – REVIEW

REVIEWER	Croston, Michelle Manchester Metropolitan University, Brooks Building, Birley Campus,
REVIEW RETURNED	24-Nov-2020

GENERAL COMMENTS	Thank you for the article it was a pleasure to review. Withing the article you clearly outline the pertaining to this issue in HIV clinical practice. The benefits to people living with HIV are clearly articulated. This is a very timely and contemporary issue within the field of HIV care.
--

REVIEWER	Tiase, Victoria University of Utah, College of Nursing
REVIEW RETURNED	10-Jan-2021

GENERAL COMMENTS	Thank you for this important contribution to patient-generated health data research. I have a few comments to improve this submission: 1. The first aim or objective mentions HIV care, but the inclusion criteria indicates management - please add detail as to how the terms were used in the screening.2. To date, there are a few other scoping reviews that examined PGHD (that you have not cited) - you may want to review them for HIV findings.3. Could you please include the search strategy per database and the dates that the searches were conducted for each?4. More detail should be added to the methods on the levels of screening. Were two levels of screening conducted with two reviewers? How was this done? How were conflicts addressed? How was the manual search of reference lists done? What that done by two people?5. Please explain the details of the thematic analysis. How was IRR determined?6. In Table 2, the studies are divided into three categories - hypothetical, prototype and pre-established - were these determined ahead of time? Please define the categories briefly.6. Data privacy/security/reliability is not mentioned in the themes. Please connect these concerns in the discussion to the results.
---

	7. It may be helpful to have a Table 1 of the studies... how many? what types? etc. 8. The abstract and conclusion do not mention that 11 articles were found - this should be front and center for readers to find. Thanks for the opportunity to review.
--	---

REVIEWER	Blondon, Katherine University Hospitals of Geneva Department of Internal Medicine, Division of General Internal Medicine
REVIEW RETURNED	21-Jan-2021

GENERAL COMMENTS	This paper is well written and satisfactory overall. I only have one point that needs clarification, and which may affect the results and discussion. My sole question resides in the criteria for excluding studies that included prompted data collection: in the definition of PG data used by the authors, there is no such limitation. PG data collecting apps and websites may provide technologies to collect all sorts of data, yet individuals with HIV may want some guidance in what kind of parameters to monitor. Healthcare providers can help orient the data collection of elements that they may need for further patient care by prompting responses to certain health issues, or with questionnaires for follow-up, symptoms, or quality of life. This can be seen in other diseases, such as diabetes, where individuals welcome the initial guidance about what to monitor, and how to document it to help understand blood glucose results. Please clarify your choice of inclusion/exclusion criteria for this study.
--

VERSION 1 – AUTHOR RESPONSE

Reviewer 2’s comments: To our knowledge, there is currently no universal definition of patient-generated data. Though several of the research studies we reviewed cited the ONC's definition of patient-generated health data, we do not believe this definition has overtly been confirmed as a gold-standard, nor are we aware if this definition has been widely ratified by the wider global digital health research community.

Given the majority of studies included within this literature review -- and indeed across the wider digital health literature -- are of US provenance, we are keen not to denigrate or ignore the ONC definition, but rather to remain cautious of prematurely adopting or implying a universal definition of PGHD until there is clearer formal consensus on the terminology within the wider research community.

Regarding to the concerns raised about discoverability, we are confident that our use of the terms 'patientgenerated health data' and 'PGHD' will be sufficient to ensure the paper's discoverability.

Reviewer 3’s comments: Thank you for your feedback that our explanation for why we have excluded prompted patient-generated data is satisfactory.

We, therefore, do not wish to submit any further revisions of the manuscript at this time.

We wish to express our gratitude to the editor, editorial staff and reviewers for their thoughtful consideration of this manuscript.

VERSION 2 – REVIEW

REVIEWER	Tiase, Victoria University of Utah, College of Nursing
REVIEW RETURNED	28-Mar-2021

GENERAL COMMENTS	Thank you for adequately addressing all of the reviewer concerns in your cover letter and the proposed changes. I just have one remaining comment - now that you have included previous work in this area and cited many more references, on page 3, would you still contend that no universal definition exists? Especially given that your references now include the use of PGHD - the accepted abbreviation for patient-generated health data - and the universal definition from ONC https://www.healthit.gov/topic/scientific-initiatives/pcor/patient-generated-health-data-pghd?? Perhaps you may want to adjust your work so it is discoverable and aligned?
---

REVIEWER	Blondon, Katherine University Hospitals of Geneva Department of Internal Medicine, Division of General Internal Medicine
REVIEW RETURNED	06-Apr-2021

GENERAL COMMENTS	The explanation about excluding prompted PG data is satisfactory. I have no further comments.
---

VERSION 2 – AUTHOR RESPONSE

Reviewer 2's comments: To our knowledge, there is currently no universal definition of patient-generated data. Though several of the research studies we reviewed cited the ONC's definition of patient-generated health data, we do not believe this definition has overtly been confirmed as a gold-standard, nor are we aware if this definition has been widely ratified by the wider global digital health research community.

Given the majority of studies included within this literature review -- and indeed across the wider digital health literature -- are of US provenance, we are keen not to denigrate or ignore the ONC definition, but rather to remain cautious of prematurely adopting or implying a universal definition of PGHD until there is clearer formal consensus on the terminology within the wider research community.

Regarding to the concerns raised about discoverability, we are confident that our use of the terms 'patientgenerated health data' and 'PGHD' will be sufficient to ensure the paper's discoverability.

Reviewer 3's comments: Thank you for your feedback that our explanation for why we have excluded prompted patient-generated data is satisfactory.

We, therefore, do not wish to submit any further revisions of the manuscript at this time.

We wish to express our gratitude to the editor, editorial staff and reviewers for their thoughtful consideration of this manuscript.